# Annual molt period and seasonal color variation in the Eared Dove´s crown

**Diego J. Valdez**[1,2]*, **Santiago M. Benitez-Vieyra**[3]

**1** Facultad de Ciencias Exactas, Físicas y Naturales, Centro de Zoología Aplicada, Universidad Nacional de Córdoba, Córdoba, Argentina, **2** Consejo Nacional de Investigaciones Científicas y Técnicas (CONICET), Instituto de Diversidad y Ecología Animal (IDEA), Córdoba, Argentina, **3** Instituto Multidisciplinario de Biología Vegetal (IMBIV), Universidad Nacional de Córdoba–Consejo Nacional de Investigaciones Científicas y Técnicas (CONICET), Córdoba, Argentina

* dvaldez@unc.edu.ar

## Abstract

Molting is an important process in which old and worn feathers are exchanged for new ones. Plumage color is determined by pigments such as carotenes, melanin and by the ultrastructure of the feather. The importance of plumage coloration has been widely studied in different groups of birds, generally at a particular time of the year. However, plumage coloration is not static and few studies have addressed the change in plumage color over time and its relationship to reproductive tasks. The Eared Dove (*Zenaida auriculata*, Des Murs, 1847) has a melanistic coloration with sexual dichromatism in different body regions. The Eared Dove´s crown is the most exposed body region during the bowing display. Our objective was therefore to accurately determine the molting period of the crown feathers and study the seasonal variation in their coloration in females and males. Our findings indicate a molting period of 6 months (January to June). The new feathers are undergoing changes in their coloration from July to December. During that period we apply an avian vision model then enabled us to reveal a seasonal variation in the coloration of the crown feathers in both sexes, as given by a change in the chromatic distances. The highest values in the chromatic distances towards the reproductive period are given by a change in the UV-violet component of the spectrum, indicating changes in the microstructure of the feather. This change in crown coloration towards the breeding season could be linked to reproductive behaviors.

## Introduction

Molt is a key physiological process in the life history of birds, in which old and worn feathers are replaced by new ones [1, 2]. Bird plumage coloration is determined by different pigments such as carotenes, melanins and psittacofulvins, or by the ultra-structure of the feather, which generates the structural colors [2, 3]. Plumage coloration has been linked to intra- and inter-specific communication processes, serving as an indicator of the "quality" of the individual for the purpose of attracting the opposite sex, as a signal to defend territory, or as camouflage [4]. Most objective studies on bird coloration focus on a particular time of the year, usually the breeding period. However, plumage coloration is not static and changes over time. Once the

**Data Availability Statement:** All relevant data are within the paper and its Supporting Information files.

**Funding:** The author(s) received no specific funding for this work.

**Competing interests:** The authors have declared that no competing interests exist.

molt is complete, the coloration of the feathers can change throughout the year due to various factors, both biotic (bacterial activity and ectoparasites) and abiotic (mechanical abrasion, dust and fat accumulation, solar irradiation, etc.) [5–9]. Variations in plumage coloration throughout the year could affect several aspects of a bird's life history both during non-breeding and breeding seasons (competition for food resources, male and female survival, mate choice, intra-sex competition, nesting sites, etc.). Several studies (mainly carried out on passerine species) have addressed seasonal variations in plumage coloration [6, 10–13]. Depending on the type of coloration (melanic, carotenic or structural coloration) a bird has, the pattern of color change throughout the year may differ according to the species [13].

In this context, the Eared Dove (*Zenaida auriculata*, Des Murs, 1847) is an opportunistic species of columbiform native to South America [14] that has a melanistic coloration type with slightly pinkish tones on the chest and belly, is gray on the back and has black spots on the face and wings [15]. Sexual dichromatism in the Eared Dove occurs in different body regions, being more evident in the crown [16], where males have around 20% more reflectance than females, with a noticeable difference in the UV region of the spectrum [16]. This is a very interesting finding since the crown is the body region most exposed during courtship [17]. The Eared Dove presents the typical bowing display common to most doves and pigeons, the male chasing the female with inflated neck and head down, thus exposing the crown, and the typical coo vocalization [17].

Although the sexual dichromatism of the Eared Dove has been addressed in detail, the studies were carried out at a specific time of the year and registered only the molting period of the primary remiges (the only molting data available at the time) [14, 16]. However, the molting period of the primary remiges is not necessarily the same as for the rest of the body, especially in the case of crown feathers.

The objective of the present study was therefore to follow the molting period of crown feathers in female and male Eared Dove and determine whether there is any seasonal variation in the coloration of these feathers that is perceptible by birds.

## Materials and methods

### Permits

The study meets Argentine legal requirements, was carried out in strict accordance with the Guidelines for Ethical Research on Laboratory and Farm Animals and Wildlife Species and had the prior approval of the ethics committee of CONICET (Resolution No. 1047 ANNEX II, 2005). The necessary permits to capture specimens of Eared Dove were provided by the Ministerio de Agua, Ambiente y Servicios Públicos de la Provincia de Córdoba, Argentina, through the Secretaria de Ambiente y Cambio Climático.

This work was carried out within the framework of a broader research where we studied the seasonal variation of sex hormones, gonadal size, and gonadal activity [18]. Bird sexing was performed by gonadal inspection [18].

### Molting period

All doves were captured monthly within the grounds of the Córdoba Zoo in Córdoba, Argentina (31˚ 25' 31.79" S 64˚ 10' 29.92" W) between March 2016 and March 2017 using a passive trap baited with commercial food for cage birds [19]. The molting period for crown feathers was determined in adult female and male Eared Doves. The crown of each dove was carefully observed under a magnifying glass to detect signs of the molting process (Fig 1A and 1B). For each bird we quantified the monthly number of molting feathers according to Rohwer and Manning (1990) [20], with molt scores as follow: 0 = no active molt, 1 = one to two growing

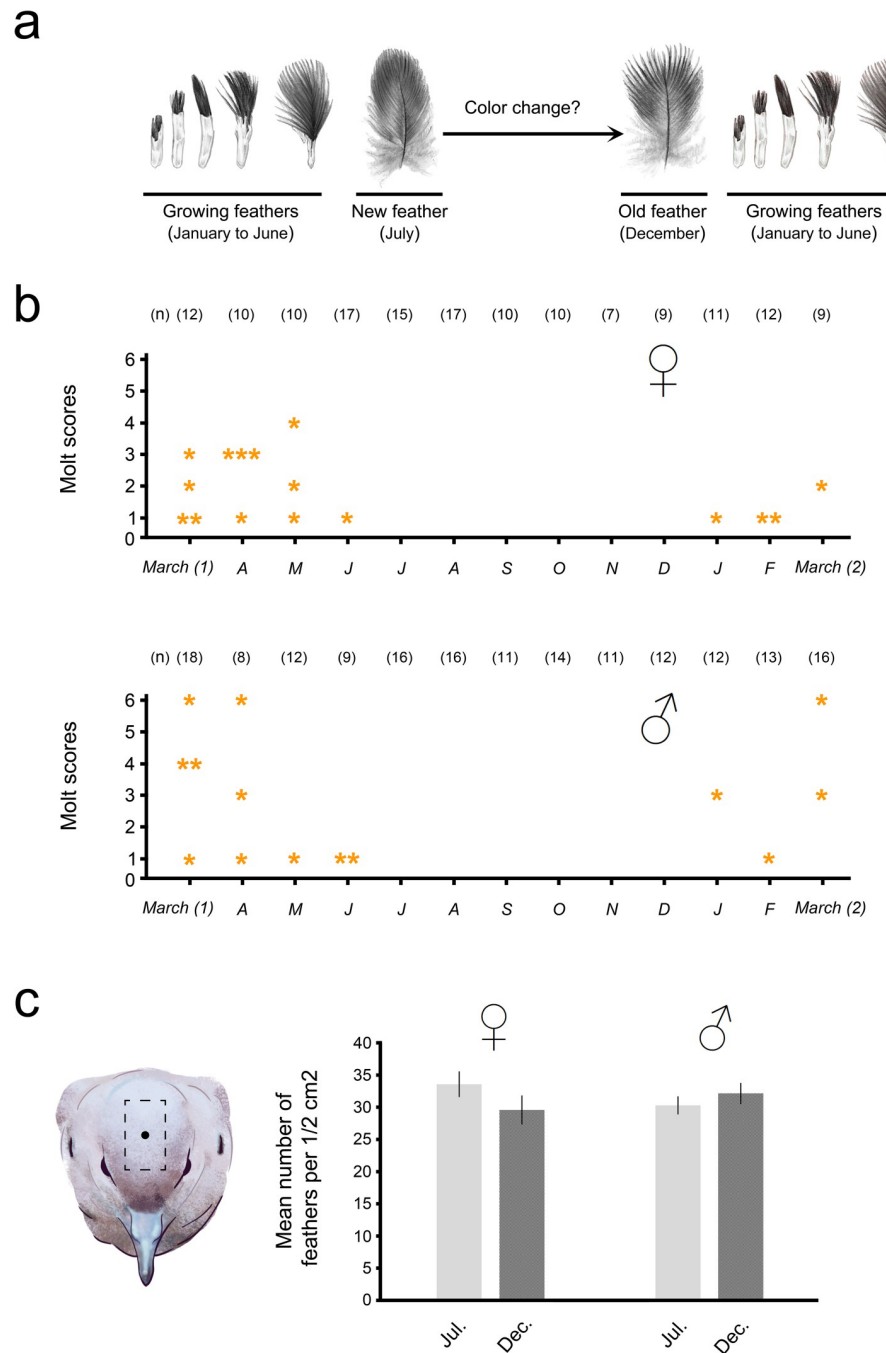

**Fig 1. Molting period of the crown in the Eared Dove. a)** Molting process of crown feathers in the Eared Dove. **b)** Molting period in both sexes of the Eared Dove and molt scores in each month, * indicates 1 bird, (n) Number of doves in each month **c)** On the left, graphic representation of the area (dotted line rectangle) of the crown where the number of feathers in females and males was measured. The dark dot in the center of the crown indicates where the spectrophotometric data was recorded. On the right, the mean number of feathers in ½ cm$^2$ of crown at two different times, winter (July) and spring (December), in both sexes. Data are expressed as mean ± SE and no significant differences were observed.

feathers, 2 = three to four growing feathers, 3 = five to six growing feathers, 4 = seven to eight growing feathers, 5 = nine to ten growing feathers and 6 = more than ten growing feathers.

## Mean number of feathers in the crown

In order to corroborate the absence of seasonal variation in the amount of feathers in the crown of both sexes, which could affect the spectrophotometric determination, the mean number of feathers in 1/2 cm$^2$ (which represents approximately 80% of the crown area) of crown area in females and males was determined at two different times of the year (Fig 1C); the first immediately after the molt, during the winter (July), and the second before the beginning of the molting period, i.e. at the end of spring (December). For this we use the same magnifying glass that we use to detect molting processes.

## Seasonal crown color variation

The reflectance data of the crown´s feathers for each dove were obtained during the months in which no molt was observed with an Ocean Optics USB4000 spectrophotometer equipped with a halogen and deuterium light source (830 Douglas Ave., Dunedin, FL, USA 34698), both connected to the sensor by a bifurcated fiberoptic cable. The plumage was illuminated and reflected light collected at 45˚ to the surface of the feather; each spectrum was the average of three readings. Reflectance data were recorded using SpectraSuite software (Ocean Optics, Inc.) and processed with the *pavo* R package [21]. The spectral analysis and avian visual model were carried out according to Valdez and Benitez-Vieyra (2016)- open access paper [16].

## Statistical analyses

We evaluated the average number of crown feathers at two different times (winter and spring) in both females and males with a *t*-test; the level of significance was ≤ 0.05. We then examined the mean (±2SE) reflectance spectra of females and males throughout the year to determine the presence of overlapping regions between months. Subsequently, we calculated all pairwise chromatic and achromatic distances (measured in JNDs) among months in females and males. For both sexes we tested whether between-month differences were greater than within-month differences using a permutational multivariate analysis of variance PERMANOVA [22], as implemented in the Adonis function of the vegan R package [23]. We then obtained bootstrap confidence intervals for the geometric means of pairwise distances between months [24] using the *bootcoldist* function of the *pavo* R package [21] with 10000 bootstrap replicates. We considered two months to be significantly different when the lower bond of the 95% confidence interval was higher than 1 JND, as this value corresponds to the discrimination limit under ideal illumination conditions [25].

All the figures were assembled with the Photoshop CS4 program.

## Results

### Sampling

We captured a total of 137 adult females and 168 adult males throughout the year, with a similar sex ratio in all months ≈1:1 (monthly details see "n" in Fig 1B).

### Molting period and mean number of feathers in the crown

The molting period for crown feathers in the Eared Dove was observed from January to June in both sexes; no molting birds were observed from July to December (Fig 1A and 1B). 50% of all molting females molted one or two feathers (molt score = 1), 18.7% three or four (molt

score = 2), 25% five or six (molt score = 3) and the remaining 6.25% seven or eight feathers (molt score = 4). Of the molting males, 42.8% molted one or two feathers (molt score = 1), 21.4% five or six (molt score = 3), 14.2% seven or eight (molt score = 4) and the remaining 21.4% more than ten feathers (molt score = 6) (Fig 1A and 1B).

Our results show that there are no significant differences in the average number of feathers between winter and spring in females or in males ($\bar{x}$ July$_{females}$ = 33.57 ±1.77; $\bar{x}$ December$_{females}$ = 29.57 ±2.02, $t_{(12)}$ = 1.488 $p$ = 0.162; $\bar{x}$ July$_{males}$ = 30.28 ±1.23; $\bar{x}$ December$_{males}$ = 32.14 ±1.43, $t_{(12)}$ = -0.98 $p$ = 0.345) (Fig 1C).

## Seasonal crown color variation

**Reflectance spectra.** A seasonal variation in the crown spectral shape for both sexes was observed, especially in the spectral region from 300 to 430 nm (UV-Violet). According to the solstice in the southern hemisphere, the spring months (October, November and December) had higher reflectance values than the winter months (July, August, September), both in females and males (Fig 2A).

## Chromatic and achromatic distances (JNDs)

In both females and males, a significant seasonal variation was only observed in the chromatic distances in *JNDs*. In contrast, no seasonal differences were observed in achromatic distances of either sex (PERMANOVA, in both cases $F_{(5,72)}$ < 2.238 and $p$ > 0.0561) throughout the entire period analyzed (Fig 2B).

The mean chromatic distances in males were higher than the discrimination threshold (1 *JND*) in November and December (last half of spring) (PERMANOVA, $F_{(5,72)}$ = 15.31 and $p$ = 0.0001), indicating males' ability to distinguish seasonal variations in crown color among themselves (Fig 2B). Females presented a significant seasonal variation in the chromatic distances, with differences between winter and the latter half of spring (PERMANOVA, $F_{(5,61)}$ = 28.26 and $p$ = 0.0001). Males presented a more limited range of variation of their chromatic distances during spring (1.2 to 2 *JNDs*) compared to females (3 to 4.5 *JNDs*), as previously reported by Valdez and Benitez-Vieyra (2016) for other body regions in this Dove.

## Discussion

In this study we report for the first time the molting period of crown feathers in the Eared Dove (perhaps the most abundant South American opportunistic Dove) and objectively determine the seasonal variation in their coloration.

## Molting period

In both females and males, the molting period of crown feathers in the Eared Dove occurs between the months of January and June (six months); no molting was observed between the months of July and December (six months). The molting pattern observed in the Eared Dove is slightly different from that described in a taxonomically related species from the Northern Hemisphere, the Mourning Dove (*Zenaida macroura*). Sullivan and Mirarchi describe that the molting of the Mouring Dove's crown feathers occurs during five months (between August and December). From January to July, no moulting was observed in this body region in both wild and captive birds [26]. This slightly shorter molting period in the Mourning Dove is perhaps due to the fact that it lives in more humid and regular environments, where the breeding and molting periods are clearly separated [26, 27]. Moreover, the molting period of crown feathers in the Eared Dove have an opposite seasonal pattern to that described by Bucher and

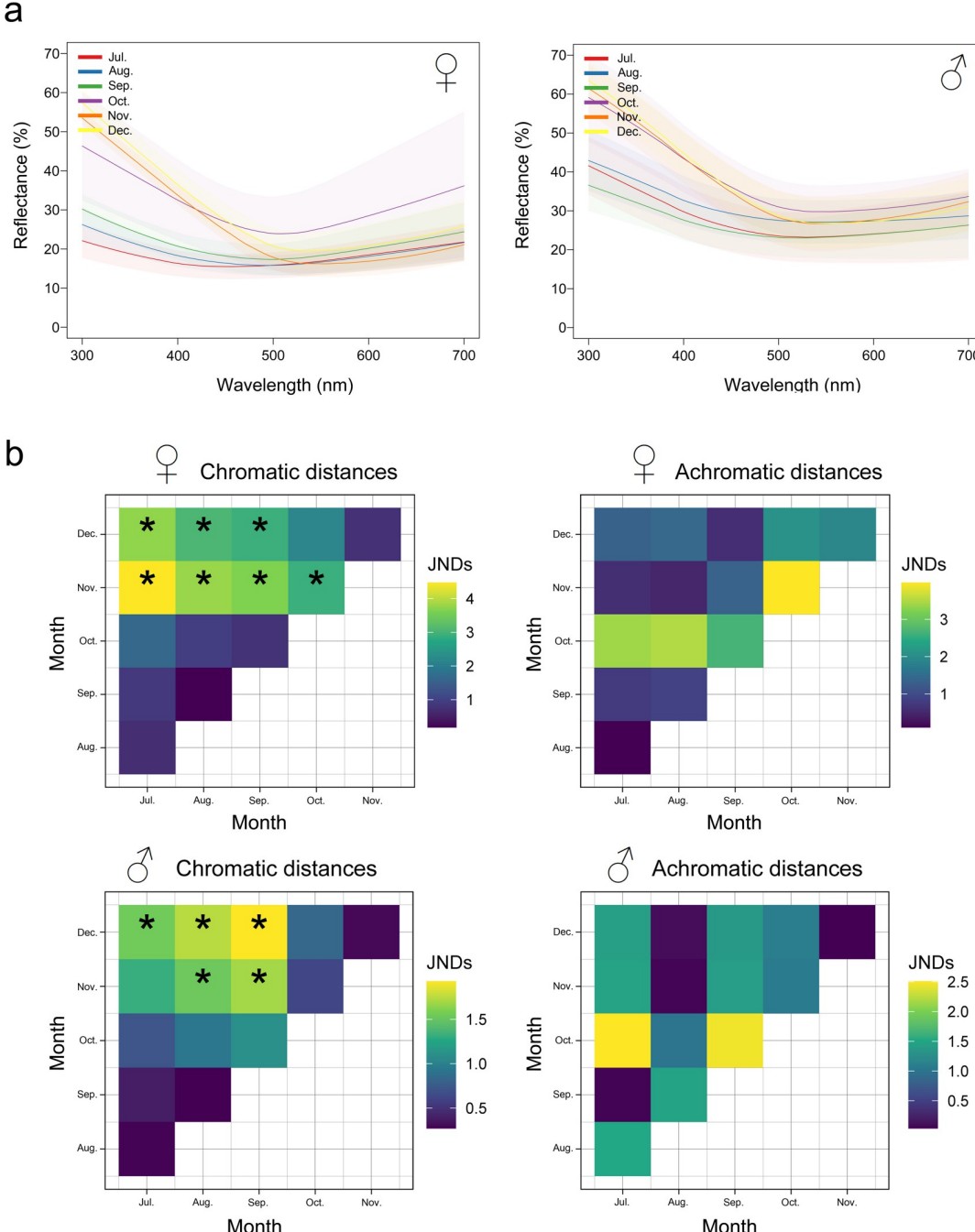

**Fig 2. Seasonal change in crown coloration. a)** Reflectance spectra of the crown of male and female Eared Doves over 6 months (winter and spring-summer) in the range visible to birds (300 to 700 nm). Data are shown as mean ± 2 SE. **b)** Chromatic and achromatic distances in *JNDs* obtained after applying an avian visual model for both sexes. Significant differences (α = 0.05; indicated by *) are only observed in the pairwise chromatic distances between winter and spring in both sexes. Month pairs were considered significantly different when the lower bond of the 95% confidence interval for the geometric mean of their distances was higher than 1 JND. Confidence intervals were obtained after 10000 bootstraps.

collaborators for the molting of primary remiges [14]. The primary remiges of the Eared Dove molt throughout the year, with a maximum turnover observed between the months of April to August. This difference in molting period could be due to the type of function that each feather

fulfills. The primary and secondary remiges are involved in flight, so they can be replaced at any time of the year, depending on the physiological conditions of the bird and external factors [28, 29]. Crown feathers on the contrary are contour feathers involved in the bowing display, so their main function is linked to coating and reproductive behavior [17, 28, 29]. Although the Eared Dove reproduces throughout the year, the greatest reproductive activity occurs from spring to early summer (September-December). The color changes displayed on the new crown feathers (from July to December) coincide with the months of greatest reproductive activity in this Dove [18, 30–33].

Using an avian visual model we observed seasonal variations in the coloration of the crown in both sexes, determined mainly by chromatic rather than achromatic distances.

## Chromatic distances

The main differences observed in the chromatic distances occur between the winter months (July, August, September) and the final months of spring / beginning of summer (November, December). Furthermore, the significant range of chromatic distances in females is greater (3 to 4.5 JNDs) than in males (1.2 to 2 JNDs). This smaller variation in the chromatic distances of males could indicate that this body region would be subject to selection pressure. These findings are in line with previous reports for this dove species and for other species of passerine birds [13, 16].

If males change crown color towards the breeding period, suggesting that they are under selection pressure, why do females also change their color from winter (non-breeding period) to spring (breeding period)? Could this region of the body also be related to the selection process in females? In other birds, it is known that as in males, the ornamentation of females is linked to individual quality and is also subject to selection pressure [34–37]. The fact that the color of the crown in female Eared Doves changes from winter to spring perhaps indicates that this region of their body could be subject to selection pressure such as sexual selection or intra-sex competitions. Seasonal changes in color variation in both sexes can also be observed in four passerine bird species such as the Robin (*Erithacus rubecula*), Blackbird (*Turdus merula*), Blue tit (*Cyanistes caeruleus*) and Great tit (*Parus major*) [10, 13]. But unlike these passerine species, the Eared Dove is an opportunistic species that successfully exploits the *ad lib* availability of food and nesting sites offered by human activities [38–41]. Our sampling data and those of Bucher et al (1977) indicate an approximate sex ratio of 1:1 throughout the year. This would allow both sexes to reduce aggression and competition behaviors between males for territory and a mate and between females for a nesting site or food sources, etc. [14, 18]. All these characteristics (opportunistic behavior, low nesting requirements and similar sex ratio) lead us to consider that perhaps the selection pressure vis-à-vis the opposite sex is relatively unimportant in this species. The seasonal change in crown coloration could be a remnant feature of a time when the species did not constitute a plague (before the 50's) and when its reproduction and feeding depended exclusively on the resources provided by natural environments (forest) without human intervention [40, 41], unlike nowadays when it inhabits urban and peri-urban areas with large extensions of crop production [40, 41]. Future experiments will be necessary to elucidate the functionality of the seasonal change in crown coloration in this dove.

It has been shown that both melanin and carotene coloration of the plumage as well as the structural coloration can change throughout the year in passerine species, although the pattern of change can be different in each species and for each type of coloration [13]. Our findings show that there is a seasonal change in the chromatic distances of both females and males. Since from 430 to 700 nm there are no significant changes in the reflectance of either sex in the months analyzed, these differences are likely due mainly to a change in the UV-violet

component of the spectrum (Fig 2A). This could indicate that the change in chromatic distances is not due to a seasonal variation in the pigment structure (changes due to photo bleaching), but rather to changes in the feather ultrastructure. This is similar to observations in the Robin, where the chromatic variation of the feathers on the back and chest increases with time, indicating an increase in reflectance at short wavelengths [13]. Contrary to the case of the Eared Dove, the chromatic variation of the black crown in the great tit is due to a decrease in UV reflectance (structural component) over time. The Great tit has high UV reflectance values just after the molt (breeding season) and low values during the non-breeding season [13]. These seasonal changes in the coloration of the plumage can be due to the accumulation of dust, fat or bacterial action on the feathers [5, 7, 8]. Although the Eared Dove and the great tit have temporally different color patterns, they coincide in having maximum values of UV reflectance during the breeding period, which, contrary to the remnant feature argument, could indicate that this portion of the electromagnetic spectrum has some importance in reproductive behavioral processes in this dove (intraspecific competition, sexual selection?). This idea is reinforced by the fact that plasma levels of testosterone in Eared Dove males are elevated during spring-summer [18]; and it is known that high testosterone levels stimulate preening behavior leading to increased UV reflectance in a passerine bird [42–44]. In the same way, it is possible that this mechanism is present in the Eared Dove.

## Achromatic distances

In at least four species of passerine birds (Robin, Blue tit, Great tit, Blackbird) seasonal variation in the color of different body regions was determined [13]. The achromatic values changed significantly throughout the year in 65% of the body regions studied, with the most significant values for structural coloration, medium vales for melanic coloration and low for carotene coloration. In these birds, achromatic values also showed sexual dichromatism [13]. On the contrary, in our study we did not observe any significant seasonal variation in the achromatic values of the crown in either female or male Eared Doves. These findings appear to be in line with those reported by Figuerola and Senar (2005) for the black crown of the Great tit, where no seasonal changes in melanic coloration from 400 to 700 nm (human visual range without UV region) were observed [11]. The lack of change in the achromatic components throughout the year in the Eared Dove is perhaps due that melanin being less sensitive to photo bleaching, suggesting that the color change is likely due mainly to structural components (UV reflectance) [45, 46]. It is not clear which are the factors affecting the structural components of the feather and therefore the UV reflectance from winter to spring in the Eared Dove, but one possible factor is the increase in microbial action during the spring-summer months due to climatic factors (warmer and more humid months) in this region [47]. It is known that bacterial action on feathers can change colorimetric parameters such as brightness, both in a species of pigeon from the northern hemisphere (*Columba livia*) and in passerine birds [7, 48–50]. But the action of the bacteria on the feathers seems to have an impact over the entire spectral range (300 to 700 nm). In our case, the spectral span from 430 to 700 nm does not change in females or males between the seasons studied and seasonal changes only occur in the UV-violet region. In this connection, a differential effect of bacterial activity in the same feather in the Pied flycatcher (*Ficedula hypoleuca*) was demonstrated. The unmelanized regions of the feather degraded more than the melanized regions, and a differential effect between females and males was observed [46]. In addition to this, Shawkey et al (2011) observed that the structural coloration of the iridescent feathers of the Mourning Dove (*Zenaida macroura*) changes with the application of cycles of hydration and dehydration [51]. In turn, Laczi et al (2021) observe changes in reflectance (UV component) in the white wing-

patch of female Collared Flycatchers (*Ficedula albicollis*) due to changes in the macrostructure of the feathers (barb angle to the rachis and vane width), possibly due to the abrasion suffered by the feathers with the nest material in the incubation stage [52].

Whether there is a differential effect of bacterial activity on female and male Eared Dove feathers (as with the Pied flycatcher) across the electromagnetic spectrum, or whether there is some other biotic or abiotic agent (ambient humidity or abrasion with nest material) that produces this effect should be addressed in future studies.

In conclusion, this is the first work to describe the molting period for the Eared Dove´s crown, which runs from mid-summer to early winter. In addition, we describe a seasonal variation of crown coloration in both sexes mainly due to variations in the UV-violet component of the spectrum, with higher values towards the reproductive season. Future studies will be necessary to elucidate the participation of UV reflectance in selection processes (between the sexes and within the gender) in this opportunistic species.

## Supporting information

**S1 File. Script that contains the descriptive, analytical and statistical analyzes carried out in the study.**
(R)

**S2 File. Individual reflectance values (females) for each of the months analyzed.**
(CSV)

**S3 File. Individual reflectance values (males) for each of the months analyzed.**
(CSV)

## Acknowledgments

A special thanks to Manuel Sosa, technical illustrator of IDEA institute, for his help in constructing Fig 1.

## Author Contributions

**Conceptualization:** Diego J. Valdez, Santiago M. Benitez-Vieyra.

**Formal analysis:** Diego J. Valdez, Santiago M. Benitez-Vieyra.

**Investigation:** Diego J. Valdez.

**Methodology:** Diego J. Valdez, Santiago M. Benitez-Vieyra.

**Resources:** Diego J. Valdez.

**Writing – original draft:** Diego J. Valdez, Santiago M. Benitez-Vieyra.

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
