## [Decision Letter · Decision Letter 0]

8 Nov 2022

PONE-D-22-25726Annual molt period and seasonal color variation in the Eared Dove´s crownPLOS ONE

Dear Dr. Valdez,

Thank you for submitting your manuscript to PLOS ONE. After careful consideration, we feel that it has merit but does not fully meet PLOS ONE’s publication criteria as it currently stands. Therefore, we invite you to submit a revised version of the manuscript that addresses the points raised during the review process.

 Please submit your revised manuscript by Dec 23 2022 11:59PM. If you will need more time than this to complete your revisions, please reply to this message or contact the journal office at plosone@plos.org. Please include the following items when submitting your revised manuscript:A rebuttal letter that responds to each point raised by the academic editor and reviewer(s). You should upload this letter as a separate file labeled 'Response to Reviewers'.A marked-up copy of your manuscript that highlights changes made to the original version. You should upload this as a separate file labeled 'Revised Manuscript with Track Changes'.An unmarked version of your revised paper without tracked changes. You should upload this as a separate file labeled 'Manuscript'.

We look forward to receiving your revised manuscript.

Kind regards,

Matthew Shawkey

Academic Editor

PLOS ONE

Journal Requirements:

Additional Editor Comments:

In my opinion this is an informative and solid descriptive paper. The reviewers have made numerous suggestions, comments, and questions that need to be addressed in the revision. I look forward to seeing the revised version. 

Reviewers' comments:

Reviewer's Responses to Questions

**Comments to the Author**

1. Is the manuscript technically sound, and do the data support the conclusions?

Reviewer #1: Partly

Reviewer #2: Yes

2. Has the statistical analysis been performed appropriately and rigorously? 

Reviewer #1: Yes

Reviewer #2: Yes

3. Have the authors made all data underlying the findings in their manuscript fully available?

Reviewer #1: Yes

Reviewer #2: Yes

4. Is the manuscript presented in an intelligible fashion and written in standard English?

Reviewer #1: Yes

Reviewer #2: Yes

5. Review Comments to the Author

Reviewer #1: PONE-D-22-25726: Annual molt period and seasonal color variation in the Eared Dove ´s crown

This paper studies the molt of crown feathers of the Eared Dove Zenaida auriculata and how feather wear affects plumage reflectance. In order to analyze the molting pattern, doves were captured throughout the year. Plumage coloration was measured during the six months without molt.

This study is descriptive and analyzes whether the expected pattern of color change due to feather abrasion and other determinants on wear is observed. There is no particular prediction on selective advantages of molting period and duration. Moreover, reproductive activity of the species in urban areas occurs throughout the year, as authors mention in line 193 (references 29 & 30), and occurs frequently during the months for which plumage color has been measured (Camargo & Araujo 2015).

Abstract:

I think the statement that feathers are exchanged for more colorful ones during the molt might be misleading. I would instead refer to molt compensating feather wear which affects plumage morphology and color.

I would also rephrase the second sentence as plumage color is fixed: “Plumage color is determined by pigments and/and plumage structure.”

It is melanic, carotenoid or structural coloration, not melanistic or carotenic (correct in abstract and main text).

Introduction:

Line 52: I think the following sentence is correct, where you mention that plumage coloration is determined by pigments and feather structure. Thus I would erase the second sentence of the paragraph.

Methods:

Doves are caught throughout the year with walk in traps and inspected for molting feathers in the crown. What are the chances of capturing the same bird and how does this affect the results?

The measurement of total feathers is not described.

Where did you measure plumage color with the spectrometer? In the lab, at site? Were individuals immediately released after measurements were taken?

How were birds sexed?

Results:

What does the number of molting feathers mean? Is it related to molting rate or what information do you extract from this data?

Reflectance spectra of figure 2 are quite dissimilar of the ones published by the authors in their paper of 2016 and this is not discussed in the paper.

Discussion:

Lines 179-185: I do not think that the molting pattern is so different from the Mourning Dove. Both occur after peak reproduction during several months (5 vs. 6 months). The fact that molt is in different months is related to the distribution of the species in different hemispheres.

Lines 191-193: If you refer to bowing display, I do not think that thermoregulation is relevant.

Lines 193-195: Why must feather replacement have taken place? If plumage reflectance is not related to feather wear or this plumage patch is not related with sexual selection?

Lines 199-204: If molting takes place during six months, why is the difference in plumage reflectance expressed in the period of one month? Nor from the reflectance spectra nor from the statistical analysis a gradual plumage color change is observed, as would be expected from feather abrasion or other factors affecting feather wear.

Lines 202-203: I do not think you can make this conclusion. It could depend on features of the feather or on other aspects that preclude feather abrasion.

Lines 207-2011: In the first place, you never showed that the coloration of the crown is under sexual selection. Second, a much more parsimonious explanation is that molt is physiologically determined and similar between sexes because of correlated physiological functions between sexes.

Line 233: How would the amount of pigments change seasonally? Pigments are deposited during feather growth.

Lines 258-262: Achromatic components are related to brightness. Amount and distribution of pigments determine feather color and overall brightness (i.e. white vs black). However, melanin pigments do not change their amount or distribution once they are deposited in the feather so you would not expect achromatic or chromatic changes related to pigments but instead related to feather microstructure that can suffer from abrasion on keratin layers or macrostructure that can be related to barbs’ and barbules’ positioning. Melanin may strengthen the structure of the barbs and barbules and decrease abrasion, but the amount or distribution in the feather will not be modified.

Reviewer #2: This study quantifies body feather moult of the eared dove and relates this to changes in the colour of crown plumage which features in sexual displays. Body feather moult is not very well understood (especially outside the northern hemisphere), and needs to be carefully quantified in the field, as the authors have done for this study. The authors show that eared doves moult their crown feathers from Jan-June, and that crown feathers have relatively lower UV reflectance in July-Sept than during the breeding season (Oct-December), suggesting UV reflectance increased after moult and that UV reflectance is highest at the height of the breeding season. This is contrary to previous studies that show decreases in UV reflectance in brightly coloured feathers over time.

I feel the paper provides a useful contribution to understanding moult timing and the relationship between moult and plumage colours under sexual selection. My main feedback is that I feel the connection between colour change of crown feathers and the timing of moult could be clarified. It would also be helpful to link the timing of moult and changes in colour by consistently referring to months and seasons in the text. I suggest more cautious language in interpreting UV reflectance as a sexual ‘signal’, it might be, but further study is needed to assess this.

Please see my detailed feedback below:

Line 26: ‘more colourful ones’ It would be good to distinguish between seasonal colour change by moult, where species alternate between a distinct breeding and non-breeding plumage, colour change by abrasion of feather tips, where feathers specially adapted to change colour (e.g. snow buntings) and colour change due to feather wear/fading and replacement with new, same-coloured but less worn feathers (as in this study).

Line 37-39: Is the change in UV due to changes in the microstructure of the feather itself or moult (new feathers) before the breeding season?

Line 56: Whether colours regularly signal individual ‘quality’ is debated in the literature, so possibly more cautious language is needed here

Line 63: Given this study is on slight colour change due to wear, it is hard to imagine this would impact survival or brooding care. I expect these kinds of effects would be more relevant for birds that undergo dramatic colour change (alternating between a distinct breeding and non-breeding plumage).

Line 110-112: It would be good to include more detail about the visual models used here, including the weber fraction used (if this applies to these models?)

Line 116-119: I suggest re-wording this sentence as it is difficult to follow

Line 153-155: Seasons are usually classed as Summer (Dec-Feb), Autumn (March-May), Winter (June-Aug) and Spring (Sept-Nov) in the southern hemisphere. I think the authors are using seasons according to the solstice, if so, it would be helpful to briefly state this. Are July-Sept the coolest months?

Line 194-195: This is difficult to relate back to the results, eared doves breed throughout the year but have greatest reproductive activity in spring (Sept-Nov) and summer (Dec-Feb), or do the authors mean from spring to early summer (Sept-Dec?). It would be helpful to include the season names and months to avoid confusion.

If breeding is mostly between Sept-Dec, then moulting is completed well ahead of breeding (by June) and birds are moulting at the end of the breeding period (late summer, Jan-Feb)? I would have thought if fresh feathers are important for breeding, then moult would be completed (crown feathers replaced) close to the start of the breeding season (see Lantz and Karubian 2016, and McQueen et al. 2021), but then the authors show UV increases after moult of the crown plumage – and suggest might be important for mate attraction – which might explain why completing moult ahead of breeding is important? This is important to clarify because I feel some of the discussion on moult and colour change is contradictory.

There are also two papers on moult and colour change that could be helpful to include here. Lantz and Karubian 2016 The Auk 133: 338-346 show that red-backed fairy-wrens re-moult their body feathers ahead of breeding, leading to an in increase in colour saturation. McQueen et al. 2021 Behavioural Ecology 32: 178-187 show that superb fairy-wrens re-moult their UV-blue crown feathers (which also feature in sexual displays) throughout the breeding season, which might explain why their UV-blue colours do not fade.

211: intra-sexual competition is also a selection pressure in itself

220: I don’t follow what the authors are saying here (which sex is the ‘opposite sex’?) Are they talking about mate choice by males and females? Do females also perform the bowing display?

245-249: An increase in UV reflectance over time contrasts with other studies and, as the authors suggest, might be explained by increased preening behaviour. There are other studies to cite that support this (see Zampiga et al. 2010 Ethology, Ecology and Evolution 16: 339-349 and Griggio et al. 2010 Behavioural Processes 84: 739-744).

286-287: I think it is overstating the results to say that the UV reflectance is an important visual signal for reproduction (just because UV reflectance is higher in the breeding season). Maybe it could be suggested as a topic for future research?

6. PLOS authors have the option to publish the peer review history of their article (what does this mean?). If published, this will include your full peer review and any attached files.

Reviewer #1: No

Reviewer #2: No

---

## [Author Response · Author response to Decision Letter 0]

19 Nov 2022

Editor Comments:

In my opinion this is an informative and solid descriptive paper. The reviewers have made numerous suggestions, comments, and questions that need to be addressed in the revision. I look forward to seeing the revised version. 

Answer 

We appreciate your opinion about our work and we have made the suggested changes in the new version of the manuscript and answered every question from the reviewers.

Review Comments to the Author

Reviewer #1: PONE-D-22-25726: Annual molt period and seasonal color variation in the Eared Dove ´s crown

This paper studies the molt of crown feathers of the Eared Dove Zenaida auriculata and how feather wear affects plumage reflectance. In order to analyze the molting pattern, doves were captured throughout the year. Plumage coloration was measured during the six months without molt.

This study is descriptive and analyzes whether the expected pattern of color change due to feather abrasion and other determinants on wear is observed. There is no particular prediction on selective advantages of molting period and duration. Moreover, reproductive activity of the species in urban areas occurs throughout the year, as authors mention in line 193 (references 29 & 30), and occurs frequently during the months for which plumage color has been measured (Camargo & Araujo 2015).

Answer

We have added to the new version of the manuscript the mentioned reference

Abstract:

I think the statement that feathers are exchanged for more colorful ones during the molt might be misleading. I would instead refer to molt compensating feather wear which affects plumage morphology and color.

I would also rephrase the second sentence as plumage color is fixed: “Plumage color is determined by pigments and/and plumage structure.”

It is melanic, carotenoid or structural coloration, not melanistic or carotenic (correct in abstract and main text).

Answer

We thank reviewer #1 for his comments. We have made the suggested changes in the new version of the manuscript.

Introduction:

Line 52: I think the following sentence is correct, where you mention that plumage coloration is determined by pigments and feather structure. Thus I would erase the second sentence of the paragraph.

Answer

We have made the suggested changes in the new version of the manuscript.

Methods:

Doves are caught throughout the year with walk in traps and inspected for molting feathers in the crown. What are the chances of capturing the same bird and how does this affect the results?

Answer

The present work was carried out within the framework of an other project where we studied the seasonal variation of sex hormones, gonadal size and gonadal activity (Maldonado et al 2020). In that published work, the birds were euthanized by decapitation (Maldonado et al 2020. Reproduction in the Eared Dove: An exception to the classic model of seasonal reproduction in birds?. Zoology 140 (2020) 125769).

We have incorporated a paragraph in the Materials and Methods section that includes this information.

We study the crown of these euthanized birds, for this reason there is no recapture.

The measurement of total feathers is not described.

Answer

We appreciate Reviewer's #1 comment. We have incorporated an explanatory paragraph in the Materials and Methods section.

Where did you measure plumage color with the spectrometer? In the lab, at site? 

Answer

The site where the birds were captured is about 50-60 meters from the laboratory of the Applied Zoology Center. In the laboratory, the samples for the previously mentioned work (Maldonado et al 2020) were processed and spectrophotometry was performed with the spectrometer described in Materials and Methods section.

Were individuals immediately released after measurements were taken?

How were birds sexed?

Answer

See previous answers

Results:

What does the number of molting feathers mean? Is it related to molting rate or what information do you extract from this data?

Answer

Figure 1b aims to show, on the one hand, the months in which we observed moult in both sexes.

On the other hand, most females have lower molting scores than males. This could indicate "physiological differences" in the molting process between males and females for this particular body región with the consequences that this could have. We have not performed any experiments to test this idea, so we decided to be cautious and not overestimate our results.

Reflectance spectra of figure 2 are quite dissimilar of the ones published by the authors in their paper of 2016 and this is not discussed in the paper.

Reviewer's #1 observation is valid, but these subtle differences may be due to the fact that the work on sexual dichromatism was carried out only during November and December, 2014 (Valdez and Benitez-Vieyra 2016).

In the current work, samples were taken for 13 months between 2016-2017 (see materials and methods section and Maldonado et al. 2020). If any environmental variable (ambient humidity, dryness, for example) differed from one sample to another and could influence the observed reflectance, we do not know.

But there is something that we do know and that is repeated from one experiment to another:

-The differences between males and females are maintained (higher reflectance values the UV-violet region in males than in females).

-The variables analyzed (chromatic and achromatic distances) have less dispersion in males than in females.

The fact that these two characteristics are repeated from one experiment to another make the results shown robust, regardless of the year, the operator, environmental conditions, etc.

Discussion:

Lines 179-185: I do not think that the molting pattern is so different from the Mourning Dove. Both occur after peak reproduction during several months (5 vs. 6 months). The fact that molt is in different months is related to the distribution of the species in different hemispheres.

Answer

We thank reviewer #1 for his comments. We have made the suggested changes in the new version of the manuscript.

Lines 191-193: If you refer to bowing display, I do not think that thermoregulation is relevant.

Answer

To avoid confusion we have eliminated the word “thermoregulation” in the new version of the manuscript.

Lines 193-195: Why must feather replacement have taken place? If plumage reflectance is not related to feather wear or this plumage patch is not related with sexual selection?

Answer

The crown is the most exposed body region during the bowing display [17], a behavior that occurs during the spring-summer seasons (greater reproductive activity) [14, 18, 30]. If these feathers are related to selection processes (female-male) or intra-gender competition processes (male-male and female-female), perhaps both males and females have to replace them before the peak of reproductive activity and said behavioral processes. (see discussion on color distances)

Lines 199-204: If molting takes place during six months, why is the difference in plumage reflectance expressed in the period of one month? Nor from the reflectance spectra nor from the statistical analysis a gradual plumage color change is observed, as would be expected from feather abrasion or other factors affecting feather wear.

Answer

Assuming that the change in reflectance (for whatever factors, abiotic or biotic) has to be gradual is a simplified appreciation of reviewer #1, more so when in other birds such as the blue tit changes in coloration from month to month also show abrupt jumps (Delhey, K., Peters, A., Johnsen, A., & Kempenaers, B. (2006). Seasonal changes in blue tit crown color: do they signal individual quality?. Behavioral Ecology, 17(5), 790- 798).

In the Eared Dove, the main changes in reflectance occur in the last months of spring and the beginning of summer (October, November and December), with the month of October being intermediate and highly variable in both sexes.

In this region (province of Córdoba, Argentina) the months of September and October are usually very variable in relation to the weather (very windy and dry, even late frosts), while the months of November and December are hot and humid. In addition to this, the local vegetation is resprouting, so there are still many tree species without leaf cover. Probably these transitional climatic variations between the cold and warm months, added to the variable vegetation cover, contribute to the observed variation in reflectance, nor can we rule out birds showing greater preening behavior for these dates.

Lines 202-203: I do not think you can make this conclusion. It could depend on features of the feather or on other aspects that preclude feather abrasion.

Answer

We appreciate reviewer's #1 comment, we have rewritten the sentence in potential form.

Lines 207-211: In the first place, you never showed that the coloration of the crown is under sexual selection. Second, a much more parsimonious explanation is that molt is physiologically determined and similar between sexes because of correlated physiological functions between sexes.

Answer

We appreciate reviewer's #1 comment, we have rewritten the sentence in potential form.

From the point of view of coverage, the crown feathers have the same function in both sexes.

From the physiological point of view, there are differences between males and females, since males have higher molting scores than females throughout the molting period (see figure 1b).

From the point of view of coloration, they clearly do not have the same function. Females can vary greatly in their coloration (from grayish brown to light gray), while males are always in the light blue range, with more or less UV reflectance (see tables 1 and 2 in Valdez and Benitez-Vieyra 2016), which "would indicate" that males could suffer selection pressure for this body region.

To think that females could also be subject to some type of selection pressure (sexual selection, intra-gender competition), is not wrong, especially when it has been observed in other bird species [34-37]. However, we have not performed any experiments to demonstrate this, so we have rewritten the sentences in potential form.

Line 233: How would the amount of pigments change seasonally? Pigments are deposited during feather growth.

Answer

Depending on the level of wear that a feather has, the air spaces (bubbles) immersed in the keratin that contain the melanin could be left open and in this way the pigment could be removed. Could this happen? Maybe yes. Has been tested? No.

To avoid confusion, we have decided to rewrite the sentence in the new version of the manuscript.

Lines 258-262: Achromatic components are related to brightness. Amount and distribution of pigments determine feather color and overall brightness (i.e. white vs black). However, melanin pigments do not change their amount or distribution once they are deposited in the feather so you would not expect achromatic or chromatic changes related to pigments but instead related to feather microstructure that can suffer from abrasion on keratin layers or macrostructure that can be related to barbs’ and barbules’ positioning. Melanin may strengthen the structure of the barbs and barbules and decrease abrasion, but the amount or distribution in the feather will not be modified.

Answer

We appreciate reviewer's #1 comment. To avoid confusion, we have decided to rewrite the sentence in the new version of the manuscript.

Reviewer #2: This study quantifies body feather moult of the eared dove and relates this to changes in the colour of crown plumage which features in sexual displays. Body feather moult is not very well understood (especially outside the northern hemisphere), and needs to be carefully quantified in the field, as the authors have done for this study. The authors show that eared doves moult their crown feathers from Jan-June, and that crown feathers have relatively lower UV reflectance in July-Sept than during the breeding season (Oct-December), suggesting UV reflectance increased after moult and that UV reflectance is highest at the height of the breeding season. This is contrary to previous studies that show decreases in UV reflectance in brightly coloured feathers over time.

I feel the paper provides a useful contribution to understanding moult timing and the relationship between moult and plumage colours under sexual selection. My main feedback is that I feel the connection between colour change of crown feathers and the timing of moult could be clarified. It would also be helpful to link the timing of moult and changes in colour by consistently referring to months and seasons in the text. I suggest more cautious language in interpreting UV reflectance as a sexual ‘signal’, it might be, but further study is needed to assess this.

Answer

We appreciate reviewer's #2 comment. We have taken the comments into account and made changes to the new version of the manuscript.

Please see my detailed feedback below:

Line 26: ‘more colourful ones’ It would be good to distinguish between seasonal colour change by moult, where species alternate between a distinct breeding and non-breeding plumage, colour change by abrasion of feather tips, where feathers specially adapted to change colour (e.g. snow buntings) and colour change due to feather wear/fading and replacement with new, same-coloured but less worn feathers (as in this study).

Answer

We appreciate Reviewer's #2 comment. Depending on the bird species and the type of feather that is studied, it will depend on the molting processes that are present. Many species have two moults in the year, one moult before the reproductive period (nuptial molt) and one after the reproductive period (postnuptial molt). The Eared Dove has an interesting molting pattern, since only one molting process (rather long) is observed in the crown that spans 6 months with different climates (winter, spring and summer). On the other hand, their primary remiges can be changed throughout the year (Bucher et al 1977), which makes this dove species have a complex moult pattern that spans different seasons.

Also see response to reviewer #1 (Abstract)

Line 37-39: Is the change in UV due to changes in the microstructure of the feather itself or moult (new feathers) before the breeding season?

Answer

There is no molt during the months in which the spectrophotometry was determined (July to December). Measuring color when the bird is molting is a serious mistake. The color changes observed in this work are mainly due to changes in the structure of the feather.

Line 56: Whether colours regularly signal individual ‘quality’ is debated in the literature, so possibly more cautious language is needed here.

Answer

We appreciate reviewer's #2 comment. We have made changes to the new version of the manuscript.

Line 63: Given this study is on slight colour change due to wear, it is hard to imagine this would impact survival or brooding care. I expect these kinds of effects would be more relevant for birds that undergo dramatic colour change (alternating between a distinct breeding and non-breeding plumage).

Answer

To avoid confusion we have decided to rewrite the sentence.

Line 110-112: It would be good to include more detail about the visual models used here, including the weber fraction used (if this applies to these models?) 

Answer

We appreciate reviewer's #2 concern. In order to avoid the manuscript being too long, all the information referring to the avian visual model used here is available in Valdez and Benitez-Vieyra 2016 published in this same open access journal.

Valdez DJ, Benitez-Vieyra SM (2016) A Spectrophotometric Study of Plumage Color in the Eared Dove (Zenaida auriculata), the Most Abundant South American Columbiforme. PLoS ONE 11(5):

e0155501. doi:10.1371/journal.pone.0155501

Data Availability Statement: Data are available from Figshare (https://dx.doi.org/10.6084/m9.figshare.3364681).

So that the reviewer can quickly read M&M, we copy the information referring to the avian visual model

“Two complementary approaches were used to determine plumage color in the Eared Dove

using the pavo package [32] of R software (R Core Team 2015 [33]): classic colorimetric variables analysis (hue, chroma and brightness) and an avian visual model. For the latter, cone

quantum catch (Q) for each of the four avian cones was calculated under a standardized daylight illumination (D65) as a representative spectrum for open habitat midday ambient light.

Although cone parameters have not been measured in Z. auriculata, the generalized spectral

cone sensitivities of VS-type avian eyes was used since this visual system characterizes all

Columbiformes studied so far [34]. The sum of the two longest-wavelength cones was used to

calculate achromatic cone stimulation.

The relative cone excitation values were then used to calculate the coordinates of body parts

in a tetrachromatic color space [35]. Finally, in order to estimate the chromatic and achromatic

contrasts among different body regions a model of avian vision was applied which assumes

that receptor noise limits discrimination in each cone [36–38]. Contrasts were characterized in

units of "just noticeable differences" (JND), such that one JND represents the threshold of possible discrimination (See S1 Text and S1 Table for more information)”.

34. Ödeen A, Håstad O. The phylogenetic distribution of ultraviolet sensitivity in birds. BMC evolutionary

biology. 2013; 13(1):36.

35. Goldsmith TH. Optimization, constraint, and history in the evolution of eyes. Quarterly Review of Biology. 1990:281–322. PMID: 2146698

36. Vorobyev M, Brandt R, Peitsch D, Laughlin SB, Menzel R. Colour thresholds and receptor noise:

behaviour and physiology compared. Vision Res. 2001; 41(5):639–53. PMID: 11226508

37. Vorobyev M, Osorio D. Receptor noise as a determinant of colour thresholds. Proc R SocLond B

BiolSci. 1998; 265(1394):351–8.

38. Vorobyev M, Osorio D, Bennett A, Marshall N, Cuthill I. Tetrachromacy, oil droplets and bird plumage

colours. J CompPhysiolA. 1998; 183(5):621–33.

Line 116-119: I suggest re-wording this sentence as it is difficult to follow.

Answer

We have made the suggested changes

Line 153-155: Seasons are usually classed as Summer (Dec-Feb), Autumn (March-May), Winter (June-Aug) and Spring (Sept-Nov) in the southern hemisphere. I think the authors are using seasons according to the solstice, if so, it would be helpful to briefly state this. Are July-Sept the coolest months?

Answer

Indeed we use the solstice to define the seasons. We have made the changes suggested in the Materials and Methods section.

In the Province of Córdoba, Argentina, the temperature begins to drop during the months of April and May, reaching the lowest values during the months of June, July and August (August is very cold, dry and windy).

Line 194-195: This is difficult to relate back to the results, eared doves breed throughout the year but have greatest reproductive activity in spring (Sept-Nov) and summer (Dec-Feb), or do the authors mean from spring to early summer (Sept-Dec?). It would be helpful to include the season names and months to avoid confusion.

Answer

We have made the suggested changes

If breeding is mostly between Sept-Dec, then moulting is completed well ahead of breeding (by June) and birds are moulting at the end of the breeding period (late summer, Jan-Feb)? 

I would have thought if fresh feathers are important for breeding, then moult would be completed (crown feathers replaced) close to the start of the breeding season (see Lantz and Karubian 2016, and McQueen et al. 2021), but then the authors show UV increases after moult of the crown plumage – and suggest might be important for mate attraction – which might explain why completing moult ahead of breeding is important? 

Answer

The molt begins in January (end of the reproductive period), continues through February, March, April, May and ends in June.

From July to December there is no molt. The new feathers of July are undergoing changes in their coloration towards December. The greatest reproductive activity of this dove occurs between the months of September and December. This is precisely what is INTERESTING. The changes suffered by the feathers are related to an increase in UV reflectance, which coincides with the months of greatest reproductive activity.

This is important to clarify because I feel some of the discussion on moult and colour change is contradictory.

Answer

This pattern of color change (increased UV reflectance), due to wear, bacterial activity and environmental factors such as ambient humidity, etc., is opposite to what is observed in other species of birds where the new, colorful feathers, with high UV reflectance, no wear, they are ready just before the reproductive period.

This is what is interesting about this work, this species of dove changes its coloration, towards the period of greatest reproductive activity, in an OTHER WAY than other species, mainly passerines (see discussion). Here the color change is due to wear (abiotic or biotic) that produce changes in the structure of the feather. In passerine species, the color change is due to molting, new and more colorful feathers.

There are also two papers on moult and colour change that could be helpful to include here. Lantz and Karubian 2016 The Auk 133: 338-346 show that red-backed fairy-wrens re-moult their body feathers ahead of breeding, leading to an in increase in colour saturation. McQueen et al. 2021 Behavioural Ecology 32: 178-187 show that superb fairy-wrens re-moult their UV-blue crown feathers (which also feature in sexual displays) throughout the breeding season, which might explain why their UV-blue colours do not fade.

Answer

We appreciate reviewer's #2 suggestion, we have incorporated the two references into the new version of the manuscript.

211: intra-sexual competition is also a selection pressure in itself

Answer

We have made changes in the new version of the manuscript

220: I don’t follow what the authors are saying here (which sex is the ‘opposite sex’?) Are they talking about mate choice by males and females? Do females also perform the bowing display?

Answer

We mean that if there were some kind of selection pressure between the sexes (from females to males and from males to females) or intra-gender, this would be relatively unimportant in this species. The changes in coloration of the crown towards the reproductive period could be a remaining characteristic when the species was not yet a pest. Currently, this species is a pest, having a similar sex ratio (1:1) and a broad opportunistic diet, so the sexes would not have strong competition to find a partner or food, nesting site etc.

That is why the sentence is written in the potential form ("perhaps").

245-249: An increase in UV reflectance over time contrasts with other studies and, as the authors suggest, might be explained by increased preening behaviour. There are other studies to cite that support this (see Zampiga et al. 2010 Ethology, Ecology and Evolution 16: 339-349 and Griggio et al. 2010 Behavioural Processes 84: 739-744).

Answer

We appreciate reviewer's #2 suggestion, we have incorporated the two suggested references.

286-287: I think it is overstating the results to say that the UV reflectance is an important visual signal for reproduction (just because UV reflectance is higher in the breeding season). Maybe it could be suggested as a topic for future research?

Answer

We appreciate reviewer's #2 comment, we have made the suggested changes in the new version of the manuscript.________________________________________

---

## [Decision Letter · Decision Letter 1]

10 Jan 2023

Annual molt period and seasonal color variation in the Eared Dove´s crown

PONE-D-22-25726R1

Dear Dr. Valdez,

We’re pleased to inform you that your manuscript has been judged scientifically suitable for publication and will be formally accepted for publication once it meets all outstanding technical requirements.

Kind regards,

Matthew Shawkey

Academic Editor

PLOS ONE

Additional Editor Comments (optional):

The authors have done a good job addressing the reviewers' comments. Please note the final suggestions of reviewer 1 (below) and incorporate if needed.

Reviewers' comments:

Reviewer's Responses to Questions

**Comments to the Author**

1. If the authors have adequately addressed your comments raised in a previous round of review and you feel that this manuscript is now acceptable for publication, you may indicate that here to bypass the “Comments to the Author” section, enter your conflict of interest statement in the “Confidential to Editor” section, and submit your "Accept" recommendation.

Reviewer #2: All comments have been addressed

2. Is the manuscript technically sound, and do the data support the conclusions?

Reviewer #2: Yes

3. Has the statistical analysis been performed appropriately and rigorously? 

Reviewer #2: Yes

4. Have the authors made all data underlying the findings in their manuscript fully available?

Reviewer #2: Yes

5. Is the manuscript presented in an intelligible fashion and written in standard English?

Reviewer #2: Yes

6. Review Comments to the Author

Reviewer #2: This paper contributes to the current understanding of moult and changes in bird plumage colours over time, which may be relevant for sexual selection. The authors have addressed my feedback, although I still have two outstanding minor suggestions that might improve the clarity of the paper.

Lines 37-39: Reading the original text, it was not clear to me whether the change in colour is because old feathers were replaced (e.g. with structurally different feathers/less worn feathers) or because of changes to old/existing feather structures (e.g. through wear/abrasion, built up of fat/preen oils), leading to colour change over time since moult. The authors could clarify this here by stating the colour change is due to changes to the same feathers over time. For example, ‘…suggesting changes in the microstructure of the feathers over time, after moult’ or something similar.

Lines 209-212 and the ‘Moulting Period’ paragraph of the discussion: I agree with the authors that the increase in UV over time after moult, coinciding with peak breeding, is interesting and a key result in their study. My original comment was about making this clear in the discussion. I found it hard to keep track of the relationship between moult timing and changes in UV (i.e. that the changes in colour occur as the feathers age over time, after the moult, which is completed well before breeding, and that old feathers displayed during peak breeding season reflect more UV). The authors explain this well in their reply to my comments and could incorporate a similar statement here. Lines 209-212 seem to imply that crown feathers are replaced during peak breeding, which is confusing as it contradicts other parts of the paper.

7. PLOS authors have the option to publish the peer review history of their article (what does this mean?). If published, this will include your full peer review and any attached files.

Reviewer #2: No

---

## [Editor Report · Acceptance letter]

15 Feb 2023

PONE-D-22-25726R1 

Annual molt period and seasonal color variation in the Eared Dove´s crown 

Dear Dr. Valdez:

I'm pleased to inform you that your manuscript has been deemed suitable for publication in PLOS ONE. Congratulations! Your manuscript is now with our production department. 

Kind regards, 

on behalf of

Dr. Matthew Shawkey 

Academic Editor

PLOS ONE